# Weakly-supervised learning for image-based classification of primary melanomas into genomic immune subgroups

**Lucy Godson**[1]                                                                    SCLG@LEEDS.AC.UK
[1] *University of Leeds*
**Navid Alemi**[1]                                                        N.ALEMIKOOHBANANI@LEEDS.AC.UK
**Jérémie Nsengimana**[2]                                        JEREMIE.NSENGIMANA@NEWCASTLE.AC.UK
[4] *Newcastle University*
**Graham P. Cook**[1]                                                               G.P.COOK@LEEDS.AC.UK
**Emily L. Clarke**[1]                                                           E.L.CLARKE@LEEDS.AC.UK
**Darren Treanor**[1]                                                            DARRENTREANOR@NHS.NET
**D. Timothy Bishop**[1]                                                         D.T.BISHOP@LEEDS.AC.UK
**Julia Newton-Bishop**[1]                                               J.A.NEWTON-BISHOP@LEEDS.AC.UK
**Ali Gooya**[1]                                                                     A.GOOYA@LEEDS.AC.UK

**Editors:** Under Review for MIDL 2022

## Abstract

Determining early-stage prognostic markers and stratifying patients for effective treatment are two key challenges for improving outcomes for melanoma patients. Previous studies have used tumour transcriptome data to stratify patients into immune subgroups, which were associated with differential melanoma specific survival and potential treatment strategies. However, acquiring transcriptome data is a time-consuming and costly process. Moreover, it is not routinely used in the current clinical workflow. Here we attempt to overcome this by developing deep learning models to classify gigapixel H&E stained pathology slides, which are well established in clinical workflows, into these immune subgroups. Previous subtyping approaches have employed supervised learning which requires fully annotated data, or have only examined single genetic mutations in melanoma patients. We leverage a multiple-instance learning approach, which only requires slide-level labels and uses an attention mechanism to highlight regions of high importance to the classification. Moreover, we show that pathology-specific self-supervised models generate better representations compared to pathology-agnostic models for improving our model performance, achieving a mean AUC of 0.76 for classifying histopathology images as high or low immune subgroups. We anticipate that this method may allow us to find new biomarkers of high importance and could act as a tool for clinicians to infer the immune landscape of tumours and stratify patients, without needing to carry out additional expensive genetic tests.

**Keywords:** digital pathology, deep learning, multiple instance learning, attention, melanoma

## 1. Introduction

Melanoma is the most aggressive form of skin cancer (NHS, 2017) and the fifth most common cancer in the UK, with a rising incidence (Cancer Research UK, 2018). One of the biggest breakthroughs in treating advanced disease is immunotherapy (Curti and Faries, 2021; Wolchok et al., 2021). Yet, the most effective and well tolerated drug, PD-1 blockade only benefits around 35% of patients (Robert et al., 2015; Ugurel et al., 2016; D. et al.,

2013). Therefore, improving our understanding of the tumour microenvironment and being able to identify disease subtypes is key to stratifying patients for treatments and improving outcomes. Through consensus clustering of patient transcriptomes, previous studies have found distinct immunological subgroups within a population ascertained cohort (the Leeds Melanoma Cohort [LMC]), with differing clinical outcomes (Nsengimana et al., 2018; Poźniak et al., 2019).

These studies carried out by the Leeds Melanoma Research group (Nsengimana et al., 2018; Poźniak et al., 2019) showed how immune gene transcriptomics could be used for finding prognostic subsets and further understanding the biological mechanisms underpinning patient immunity. Most patients in the LMC were enrolled before immunotherapeutic treatments were established, meaning there is no treatment data. Nevertheless, the identified immune subgroups had distinctive features that are targeted by multiple lines of immunotherapies currently in use, in development or in clinical trials such as immune checkpoint molecules (Gibney et al., 2016; Snyder et al., 2014; Acharya et al., 2020). Moreover these subgroups had added prognostic value over the 8th American Joint Committee on Cancer (AJCC) staging system, which is the current system used for devising prognostic groups for primary melanoma patients (Gershenwald and Scolyer, 2018).

While this kind of patient stratification potentially offers early detection of prognostic biomarkers, clustering analysis of tumour transcriptome data is not routinely carried out for melanoma patients in a clinical setting and can be expensive. In recent years, the digitisation of tumour whole slide images (WSIs) and the development of convolutional neural networks (CNN), has led to the subtyping of tumours into genetic groups, through identifying the corresponding morphological patterns in the WSIs (Coudray et al., 2018; Mobadersany et al., 2018; Kim et al., 2020; Kather et al., 2020). Moreover a recent study by Schmauch et al. (2020) showed how neural networks have the propensity to learn immunological features from histopathology images which correlate with genetic expression of certain immune cell signatures. However, to our knowledge, these models have not been used to subtype patients into immune subgroups based on genetic signatures.

Overall our contribution is threefold: (1) We are the first group to develop a model to classify H&E images of melanoma into immune based genomic subgroups. (2) We show that a pathology-specific feature extractor improves model performance. (3) We show how attention-based heatmaps can be used to identify prognostic biomarkers for these subgroups.

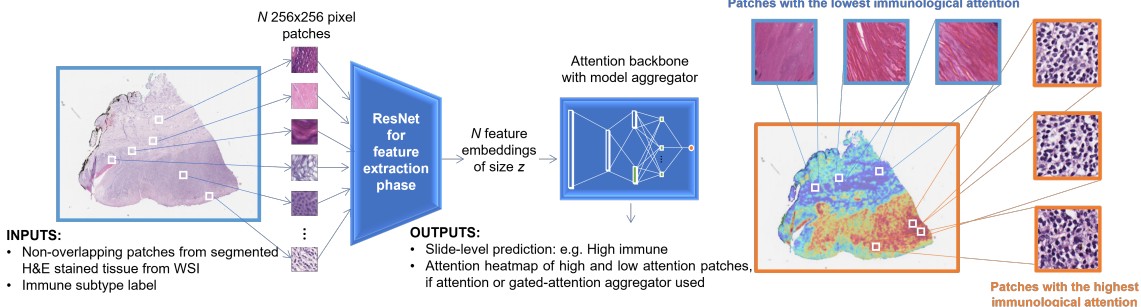

Figure 1: Overview of methods.

## 2. Related Works

**Multiple Instance Learning.** A major challenge when analysing WSIs, is finding a computationally feasible way to analyse gigapixel images. This challenge is further complicated by the fact diagnostically important regions of interest (ROI) can take up only a small fraction of the image. Furthermore, pathologists do not routinely perform pixel-level annotations on ROI, so many datasets only have a slide-level label for the WSI. One methodology that has been implemented in an attempt to resolve these challenges, is multiple instance learning (MIL) (Dietterich et al., 1997; Maron and Lozano-Pérez, 1998). MIL is a weakly supervised method that treats the image as a 'bag' made up of 'instances' which are the patches. Allowing us to solve the diagnostic task of predicting the slide-level label, whilst also flagging patches within the WSI that have triggered the slide-level label prediction.

**Attention with Multiple Instance Learning.** While initial studies applying MIL for pathology slide classification tasks performed well (Campanella et al., 2019, 2018) they required large amounts of data, or used max-pooling and averaging aggregator functions, which were pre-defined, non-trainable and lacked interpretability. Ilse et al. (2018) developed an attention-based MIL pooling function, which uses an average of weighted instances, and uses a neural network to determine the weights for each instance. By using a weighted aggregation function, it is possible to also determine which instances contribute the most to the slide-level prediction. This is important for model interpretability, which is especially pertinent in the medical field, where an incorrect diagnosis could be extremely harmful.

**Subtyping histopathology images.** Following this Lu et al. (2020). extended this attention-based methodology with an instance clustering task, to improve instance labelling accuracy when subtyping renal cell carcinoma and non-small-cell lung from histopathology images. Previous methodologies, have used feature extractors trained on the benchmark visual recognition dataset ImageNet (Russakovsky et al., 2015), which is comprised of millions of annotated images, from thousands of categories, such as food, locations, animals and people. While using a pretrained network can reduce computational cost and extract useful features from a wide array of images, it may also reduce classification accuracy for certain tasks. Yu et al. (2020), reported that when detecting TP53 mutational status from Haematoxylin and Eosin (H&E) stained images, the classification is based on pixel intensity within the cytoplasm, which may not be well represented in nonhistological image datasets like ImageNet. Furthermore Noorbakhsh et al. (2020) found that network AUC improved when training CNN parameters on cancer images and noted that pre-training with cancer histology images may improve distinction between classes where differences are more subtle. Moreover, Saillard et al. (2021) further improved upon these methodologies by implementing a self-supervised learning (SSL) MoCo V2 model, which had been trained using histopathology images, to extract feature embeddings. In our paper we extend these methodologies to our novel problem and compare how using a SSL ResNet18 (Ciga et al., 2021) can improve classification of immune subtypes using histopathology images.

## 3. Dataset

The primary dataset used to develop the models is from the LMC (Appendix A). This dataset includes 667 digitised WSIs of patient tumours with corresponding immune subgroup labels. All slides come from Formalin-Fixed Paraffin-Embedded (FFPE) blocks and

were stained using H&E. WSIs were scanned in batches using a Leica Biosystems Aperio Digital Pathology Slide Scanner, at 0.25 micrometers-per-pixel (m.p.p.). The tumour transcriptomic data that was used to develop the immune subgroup labels was produced from the archived FFPE tumour blocks, using Illumina array DASL HT12.4 and normalised using standard methods as described in the study by Nsengimana et al. (2018).

## 4. Methods

### 4.1. Segmentation and feature extraction

The H&E tissue in the WSI was segmented from the background using the thresholding and morphological operations described by Lu et al. (2020). The tissue from the segmented images was then split into 256 pixel x 256 pixel non-overlapping patches at three different magnifications (10x, 20x and 40x). These patches were initially used as inputs for a ResNet50 CNN architecture, pretrained using the ImageNet dataset (Russakovsky et al., 2015), which extracted 1024-dimensional feature embeddings from each patch. We then experimented with using a SSL ResNet18 from Ciga et al. (2021), which had been pretrained using histopathology images. This CNN architecture extracted 512-dimensional feature embeddings from the patches.

### 4.2. Model architectures and training

ResNet50 CNN representations were then further compressed by a fully connected (FC) layer, to 512-dimensional vectors $h_k$ (the ResNet18 features were already this size). We then tested different MIL pooling aggregation functions for classifying the immune subtypes from the patch embeddings.

The baseline methodology we used was a max-pooling MIL method:

$$\forall_{m=1,...,M} : z_m = \max_{k=1,...,K}(h_{km}),$$

where $z_m$ is the slide-level representation for the $m^{th}$ slide from the $M$ WSIs and $h_{km}$ is the low dimensional patch representation with the highest probability for a certain class that provides the overall WSI label.

Both of the attention-based MIL pooling functions with and without gating (Ilse et al., 2018), calculate $z$, the slide-level representation, through the aggregation of the patch feature embeddings and corresponding weights:

$$z = \sum_{k=1}^{K} a_k h_k,$$

where $h_k$ are the $K$ patch embeddings and $a_k$ are the weights that are derived from the neural network's attention backbone. The attention weights ($a_k$) add to one to be invariant of the number of patch embeddings in a slide. The weights for the attention mechanism without gating are formulated by:

$$a_k = \frac{\exp\left(\boldsymbol{w^T} \tanh(\boldsymbol{V} \boldsymbol{h_k^T})\right)}{\sum_{j=1}^{K} \exp\left(\boldsymbol{w^T} \tanh(\boldsymbol{V} \boldsymbol{h_j^T})\right)},$$

where $\mathbf{w} \in \mathbb{R}^{1024 \times 512}$ and $\mathbf{V} \in \mathbb{R}^{512 \times 256}$ are the first and second FC layers of the neural network, respectively. Hyperbolic *tanh* element-wise operations allow for gradient flow of both positive and negative values. Moreover, we implemented a gated attention mechanism, developed by Dauphin et al. (2017) to introduce sigmoid non-linearity for learning more complex relationships:

$$a_k = \frac{\exp(\boldsymbol{w^T} \tanh(\boldsymbol{Vh_k^T})) \odot sigm(\boldsymbol{Uh_k^T})}{\sum_{j=1}^{K} \exp(\boldsymbol{w^T} \tanh(\boldsymbol{Vh_j^T})) \odot sigm(\boldsymbol{Uh_k^T})},$$

where $\mathbf{U} \in \mathbb{R}^{512 \times 256}$ and $\mathbf{V} \in \mathbb{R}^{512 \times 256}$ are stacked FC layers parameterised by the network and form the attention backbone and $\odot$ refers to element-wise multiplications. Following the stacked FC layers, each class has a parallel attention branch, with an attention-based pooling function. The values from the aggregators are then used as inputs for a softmax function, which determines the overall slide-level prediction. A schematic of the attention-based architectures are shown in Figure 3 in the Appendix.

Furthermore, we also used an instance-level clustering mechanism developed by Lu et al. (2020). This mechanism was developed to improve feature learning between classes, by using pseudolabels, with a support vector machine (SVM) loss function to increase separation between the highly and least attended patches within an image (Lu et al., 2020).

The networks were trained using a cross-entropy loss function, comparing the slide label with the predicted slide-level label, to derive the parameters. A learning rate of $2 \times 10e^{-4}$, weight decay of $1 \times 10e^{-5}$. Dropout with a probability of 0.25 was used after each layer of the attention backbone and after the FC layer in the max-MIL model. Moreover, we trained models using the 3 immune subgroups found by Poźniak et al. (2019) and also looked at training models using only the "high immune" and "low immune" subgroups. The datasets were split with 80% data being used for training data and 10% being kept for both the test and validation datasets (Table 4 in Appendix). Furthermore, during training, to mitigate class imbalances between subtypes within the training data, a slide is sampled proportion to the inverse of the frequency of its ground truth class. The models were trained for a minimum of 50 epochs, with early stopping if the validation loss did not improve for 2 epochs continuously, to prevent overfitting.

### 4.3. Visual attention maps

Visual attention maps were developed by using the attention weight scores $(a_k)$ for the patch embeddings. The attention scores were converted to percentages, with the highest (1.0), being the most highly attended patch for the predicted slide-level label and the lowest (0.0) being the least attended patch for the predicted slide-level label. These percentage scores were then converted to an red green blue (RGB) colourmap which is overlaid over the WSI, with red tiles indicating highly attended patches and blue tiles indicating low attention, which contribute less to the subtype label prediction.

### 5. Results

Initial experiments were carried out by training models using the 3 subgroups determined by Poźniak et al. (2019). The three subgroups are the "high immune" class, which corresponds

to patients with a greater inferred immune cell infiltration in the primary tumour and better associated patient survival outcomes, the "intermediate immune" class which corresponds to less inferred immune cell infiltrate in the primary tumour and the "low immune" class which had the least inferred immune cell infiltrate in the tumour and worst survival response of patients. Therefore we worked under the assumption that each group with a distinct immune genetic signature would have a distinct histological pattern, that could be determined using a ResNet convolutional neural network for feature extraction.

Table 1: Mean area under the curve (AUC) and accuracy metrics ± standard deviation for the test set using 10-fold cross validation. The models were trained with 40x magnification patches from "high", "intermediate", and "low immune" classes. The highest mean AUC and accuracies are highlighted in bold.

| Method | AUC | Accuracy |
|---|---|---|
| Gated-Attention | **0.60(± 0.05)** | 0.41(± 0.06) |
| Attention | 0.58(± 0.06) | **0.43(± 0.07)** |
| Gated-Attention with clustering | 0.58(± 0.07) | **0.43(± 0.07)** |
| Attention with clustering | 0.56(± 0.05) | 0.41(± 0.04) |
| Max-pooling MIL | 0.54(± 0.03) | 0.39(± 0.04) |

We found the Gated-attention model developed by Ilse et al. (2018) gave the best mean AUC performance of 0.60. However, as the performance for all models was quite low, we then trained the models using only the "high" and "low immune" subgroup labelled images, as we believed these images were more likely to have discriminable features, due to inferred immune infiltrate within the tumour being much more polarised, compared to the intermediate subtype (Table 2). Furthermore we experimented with 3 different patch magnifications to determine the effect that the level of detail within patches would have on model performance with 20x and 40x shown in Table 2 and x10 shown in Table 5 in the Appendix. We also compared whether using a ResNet18 trained by SSL using histopathology images (Ciga et al., 2021), would improve on classifications using feature embeddings learned using a ResNet50 pretrained with ImageNet images.

Overall, using the highest resolution 40x patches, appeared to generate the best performance, with both the Gated-Attention and Attention models achieving a mean AUC of 0.73 from 10-fold cross validation. Moreover the mean AUC of all models, apart from the Max-pooling MIL model, improved when training networks with only the low and high subtype WSIs. We also saw an overall improvement in mean AUC for all models using feature representations learnt from 20x patches using the SSL ResNet18, apart from the Max-pooling MIL baseline model (Table 2). Whereas only the models using attention without gating improved when using a SSL ResNet18 for feature extraction with 40x patches (Table 2). However, the Attention model without gating trained with 40x patches, using the SSL ResNet18, achieved the highest mean AUC of 0.76 with a standard deviation of 0.05 for detecting high and low immune subtypes from histopathology images (Table 2).

Furthermore, we developed attention heatmaps, for the attention models, showing where the patches with the highest attention weights were located (Figure 2). The heatmaps help

Table 2: Mean area under the curve (AUC) and accuracy metrics ± standard deviation for models on the test set using 10-fold cross validation. The models were trained with 20x and 40x magnification patches from "high immune" and "low immune" classes. Furthermore we compare using input features from a pathology-agnostic and pathology-specific model. The highest mean AUC are highlighted in bold.

| | 20x | | 40x | |
|---|---|---|---|---|
| Method | ImageNet | SSL Histo | ImageNet | SSL Histo |
| Gated-Attention | 0.62(± 0.11) | **0.75(± 0.08)** | **0.73(± 0.09)** | 0.69(± 0.07) |
| Attention | **0.69(± 0.08)** | **0.75(± 0.07)** | **0.73(± 0.08)** | **0.76(± 0.05)** |
| Gated-attention with clustering | 0.59(± 0.08) | 0.73(± 0.09) | 0.59(± 0.10) | 0.67(± 0.12) |
| Attention with clustering | 0.67(± 0.07) | **0.75(± 0.06)** | 0.66(± 0.08) | 0.73(± 0.09) |
| Max-pooling MIL | 0.66(± 0.11) | 0.54(± 0.09) | 0.66(± 0.01) | 0.48(± 0.08) |

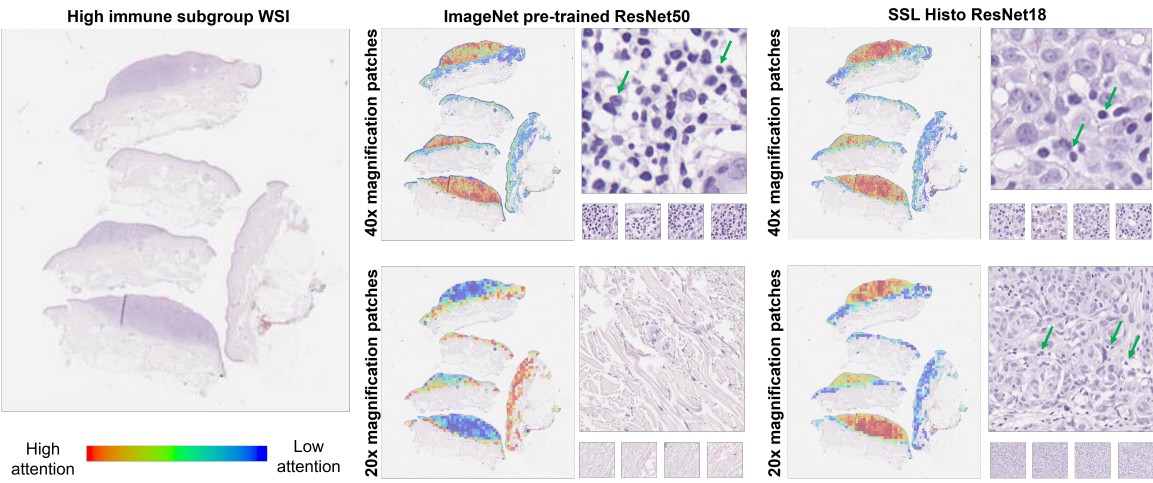

Figure 2: Attention heatmaps generated from the gated-attention models, with the five patches that contributed the most to the "high immune" subgroup prediction.

to demonstrate how the feature extraction method can alter which features and therefore regions contribute the most to the model's classification. For both magnifications, the models that use features from the SSL ResNet18 pretrained with histopathology images, have high attention regions concentrated in the darker tumour regions of the WSI (the red areas in Figure 2). Furthermore these regions contain tumour infiltrating lymphocytes (TILs), the dark circular cells indicated by the green arrows in the patches. Whereas, the high attention 20x patches, for the model using ImageNet derived features do not contain tumour or immune cells and contain regions of reticular dermis, which is not seen as a prognostic marker for melanoma patients. Moreover, the 40x patches and attention maps

are much more similar, which aligns with the similar mean AUC scores seen when altering the feature extraction step for this resolution (Table 2).

## 6. Discussion and conclusion

Recent studies (Poźniak et al., 2019; Nsengimana et al., 2018) have shown that melanoma patients can be stratified into subgroups, with added prognostic value compared to the current melanoma staging system (Gershenwald and Scolyer, 2018). However, these studies are mainly carried out using transcriptomic data, which can be expensive and time consuming. Here we show that routinely used H&E images can be used to develop models that classify patients into these subgroups. Nevertheless, deciphering the "intermediate" subgroup remains a challenge, due to the tumour heterogeneity and complexity within this group. To tackle this problem we may need to look at further dividing this subgroup, as a previous study by Nsengimana et al. (2018) found two distinct subgroups which overlap with this "intermediate" group, or use techniques to learn more discriminant representations of the images, such as deep Fisher Discriminant analysis (Díaz-Vico and Dorronsoro, 2020).

Moreover our results add to the evidence (Saillard et al., 2021; Yu et al., 2020; Noorbakhsh et al., 2020) that the feature extraction methodology can be important for capturing subtle differences in biological subtypes. The attention maps for 20x patches show that the ImageNet ResNet50 features can be acellular, focusing on regions not seen as prognostically significant (Figure 2).Whereas the high attention patches, from utilising the pathology-specific ResNet18 were concentrated in regions containing tumour and TILs, which are prognostically significant (Schatton et al., 2014; Fu et al., 2019). This demonstrates that pathology-specific feature extraction can improve feature learning in tumour regions, with 20x patches providing a balance of lower level immune cell features and higher level tissue architecture. Whereas the ImageNet pretrained models perform better when trained with 40x magnification patches, suggesting they are less able to learn contextual features, focusing on immune cells. However, in order to implement this tool in the clinical management of melanoma, performance would need to be further improved. This may involve training the feature extractor with more domain specific melanoma images, or adapting our pipeline to include a transformer architecture (Dosovitskiy et al., 2021; Wang et al., 2021) to learn non-local context, when using patch-level inputs. On the other hand, a graph-based method similar to the approach by Failmezger et al. (2020) could be used to model spatial configurations, levels of infiltration and interactions of immune cells in the melanoma microenvironment, or graph convolutional neural networks (Chen et al., 2020) could be used to examine morphometric features of cells from histopathology images, improving subgroup classification. In addition, our results are based on one dataset from a population-based cohort, therefore rigorous validation using large datasets from different centres would need to be carried out in order to assess the transferability of our results.

To the best of our knowledge this is one of the first studies that attempts to subtype melanoma histopathology images based on immune genetic signatures from transcriptomic data. We believe with improvements to the model architecture, this could be a helpful tool for pathologists, to flag potential prognostic biomarkers and aid treatment decisions, without the need for additional genetic tests.

## Acknowledgments

This work was supported by the Engineering and Physical Sciences Research Council (EP-SRC) [EP/S024336/1] and funded by Cancer Research UK [C588/A19167, C8216/A6129, and C588/A10721 and NIH CA83115]. We would like to thank the Research Computing team at Leeds, who have been really helpful when we made use of the University of Leeds high performance computing services to gain GPU access through ARC4. Furthermore this work made use of the facilities of the N8 Centre of Excellence in Computationally Intensive Research (N8 CIR) provided and funded by the N8 research partnership and EPSRC [EP/T022167/1]. The Centre is co-ordinated by the Universities of Durham, Manchester and York.

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

## Appendix A.  Description of Leeds Melanoma Cohort.

The Leeds Melanoma Cohort (LMC) is population-based study in the North of England by the Leeds Melanoma Research Group led by Prof. Julia Newton-Bishop, Consulant Dermatologist. Recruitment of melanoma patients began in 2000 and follow up has continued since then. The cohort recruitment received ethical approval MREC 1/03/57 and PIAG3-09(d)/2003. The transcriptomic data used to generate the immune subgroup labels was deposited into the European Genome Phenome Archive public repository with accession number EGAS00001002922. The imaging data is from Virtual Pathology at the University of Leeds (https://www.virtualpathology.leeds.ac.uk/). To request access to the raw data please contact the Virtual Pathology team.

## Appendix B.  Code availability

The source code of this work is available at: https://github.com/lucyOCg/Weakly-supervised-learning-for-histopathology-image-based-classification.

## Appendix C.  Average number of patches extracted per slide for the different resolutions.

Table 3: Average number of patches extracted per slide for the different resolutions for the three different resolutions.

| Resolution | Average number of patches per slide |
| --- | --- |
| 10x | 133 |
| 20x | 5258 |
| 40x | 22496 |

## Appendix D.  Splits of the dataset

Table 4: Dataset splits, showing the number of digitised histopathology images labelled with the 3 different immune subgroups found by Poźniak et al. (2019), within the training, validation and test sets.

| Immune subtype | Train | Validation | Test |
| --- | --- | --- | --- |
| Low | 204 | 26 | 26 |
| Intermediate | 209 | 26 | 26 |
| High | 120 | 15 | 15 |

## Appendix E. Attention-based MIL architecture.

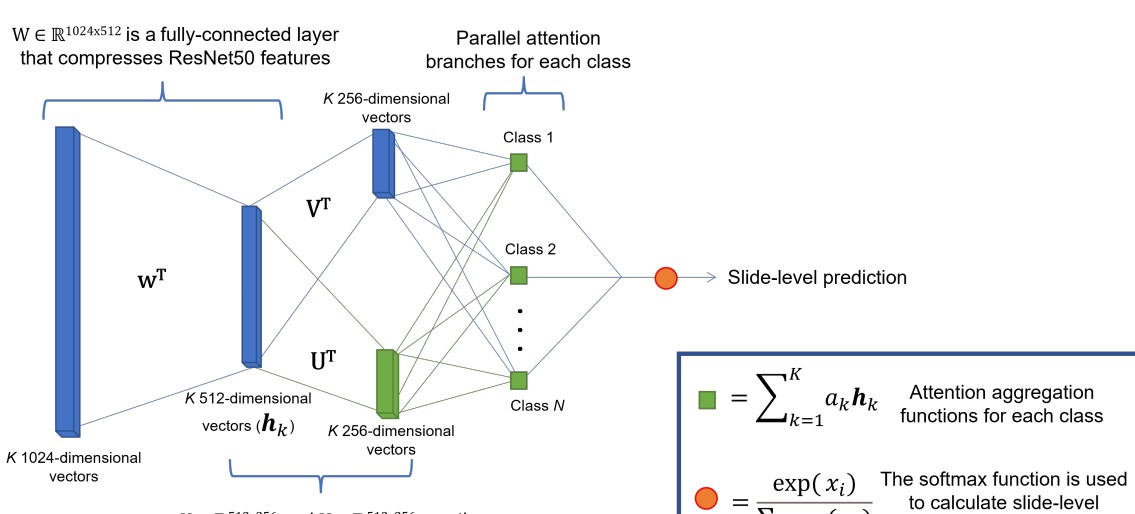

Figure 3: Network architecture for models that use the gated-attention and attention aggregation functions. The SSL ResNet18 features do not require the first fully-connected layer for compression to 512-dimensions.

## Appendix F. Table for experiments carried out using 10x patches.

Table 5: Mean area under the curve (AUC) and accuracy with standard deviation for the test set using 10-fold cross validation. The models were trained with features from x10 magnification patches from high and low immune labelled subtypes extracted by either a ResNet50 pretrained on the ImageNet dataset, or a ResNet18 pretrained using self supervised learning with digital histopathology images. The highest mean AUC and accuracy from the models are highlighted in bold.

| | AUC | | Accuracy | |
| --- | --- | --- | --- | --- |
| Method | ImageNet | SSL Histo | ImageNet | SSL Histo |
| Gated-Attention | **0.58($\pm$ 0.10)** | 0.58($\pm$ 0.09) | **0.62($\pm$ 0.07)** | **0.63($\pm$ 0.08)** |
| Attention | **0.58($\pm$ 0.10)** | 0.58($\pm$ 0.06) | 0.61($\pm$ 0.07) | 0.61($\pm$ 0.05) |
| Gated-attention with clustering | 0.57($\pm$ 0.09) | **0.60($\pm$ 0.09)** | **0.62($\pm$ 0.05)** | **0.63($\pm$ 0.04)** |
| Attention with clustering | 0.57($\pm$ 0.09) | 0.58($\pm$ 0.07) | 0.61($\pm$ 0.06) | 0.61($\pm$ 0.06) |
| Max-pooling MIL | 0.52($\pm$ 0.08) | 0.53($\pm$ 0.06) | 0.60($\pm$ 0.05) | 0.60($\pm$ 0.06) |

## Appendix G. Table for the accuracy of models trained with 20x and 40x magnification patches.

Table 6: Mean accuracy and standard deviation for the test set using 10-fold cross validation. The models were trained with features from 20x and 40x magnification patches from high and low immune labelled subtypes extracted by either a ResNet50 pretrained on the ImageNet dataset, or a ResNet18 pretrained using self supervised learning with digital histopathology images. The highest mean accuracies from the models are highlighted in bold.

| Method | 20x | | 40x | |
|---|---|---|---|---|
| | ImageNet | SSL Histo | ImageNet | SSL Histo |
| Gated-Attention | 0.66($\pm$ 0.07) | 0.71($\pm$ 0.07) | **0.70($\pm$ 0.07)** | 0.66($\pm$ 0.08) |
| Attention | **0.67($\pm$ 0.07)** | 0.70($\pm$ 0.06) | **0.70($\pm$ 0.07)** | **0.71($\pm$ 0.05)** |
| Gated-attention with clustering | 0.64($\pm$ 0.06) | **0.73($\pm$ 0.05)** | 0.61($\pm$ 0.10) | 0.65($\pm$ 0.07) |
| Attention with clustering | 0.62($\pm$ 0.05) | 0.68($\pm$ 0.08) | 0.64($\pm$ 0.07) | 0.70($\pm$ 0.08) |
| Max-pooling MIL | 0.68($\pm$ 0.04) | 0.61($\pm$ 0.05) | 0.68($\pm$ 0.08) | 0.60($\pm$ 0.06) |

## Appendix H. Data analysis

The analysis has been implemented using Python (version 3.7.5) and Pytorch (version 1.3.1) deep-learning library to experiment with models.

## Appendix I. High and Low immune subtype attention-based heatmaps with high attention patches

The pathology-specific model focuses in cellular regions, with more high attention (red) patches localised in the tumour region of the tissue. The pathology-specific model focuses in cellular regions, with more high attention (red) patches localised in the tumour region of the tissue. Whereas the pathology-agnostic region focuses on inert acellular regions, such as reticular dermis (Figure 4.A) and areas of keratin (Figure 5.A).

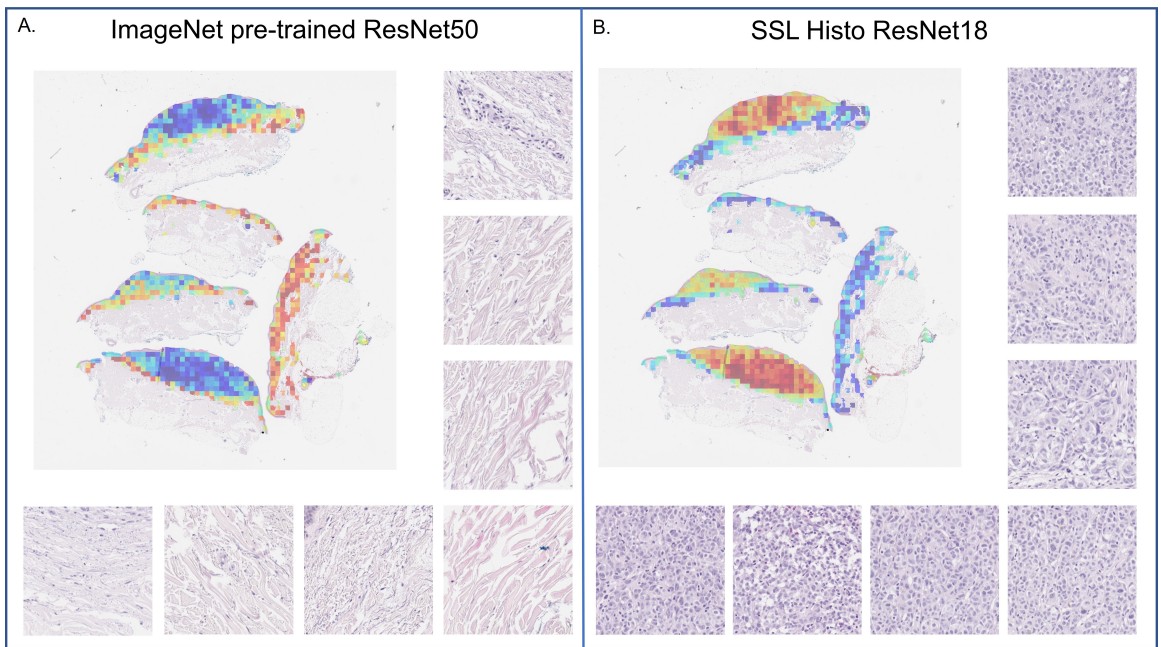

Figure 4: Attention-based heatmaps from using 20x magnification patches with different feature extraction methods for the high immune subtype. Each heatmap is next to seven of the patches with highest immunological attention scores for each method. (A.) The patch representations were derived using a pathology-agnostic ResNet50 pretrained using ImageNet. (B.) The patch representations were derived using a pathology-specific ResNet18 pretrained using self-supervised learning on histopathology images.

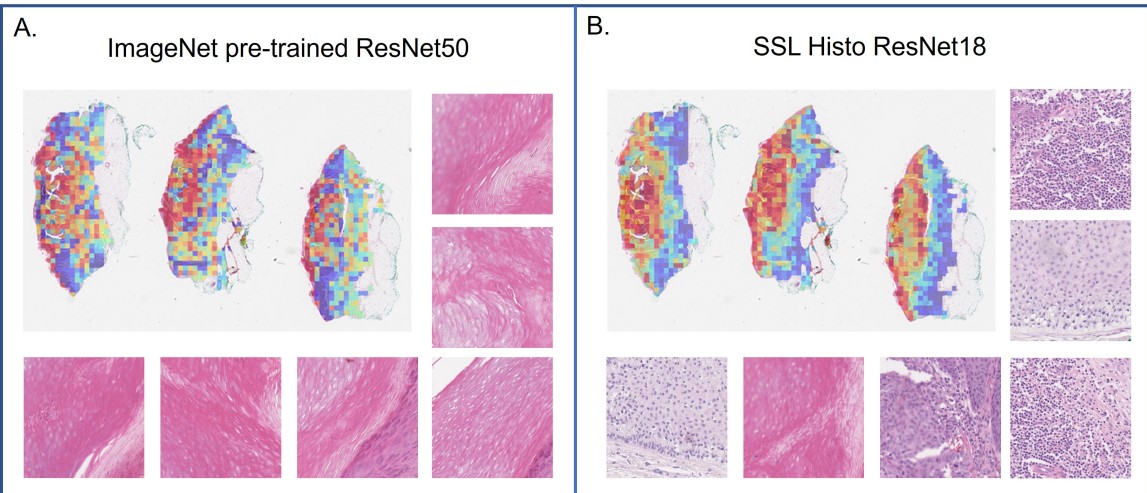

Figure 5: Attention-based heatmaps from using 20x magnification patches with different feature extraction methods for the low immune subtype. Each heatmap is next to six of the patches with highest immunological attention scores for each method. (A.) The patch representations were derived using a pathology-agnostic ResNet50 pretrained using ImageNet. (B.) The patch representations were derived using a pathology-specific ResNet18 pretrained using self-supervised learning on histopathology images.

