# OpenReview forum: "Weakly-supervised learning for image-based classification of primary melanomas into genomic immune subgroups"
_MIDL.io/2022/Conference — MIDL 2022_

### Official Review · Reviewer_tZcB · 2022-01-21

**Confidence:** 4
**Preliminary Rating:** 3
**Recommendation:** Poster

**Summary:**

This work aims to solve the weakly-supervised problem on a whole slide image application, that uses multiple instance learning to classify primary melanomas into genomic immune subgroups. The experimental results demonstrate its feasibility, as it reaches a mean AUC value of 0.76 for classifying histopathology images as high or low immune subgroups.

**Strengths:**

* It is the first work to apply multiple instance learning to this specific clinical problem based on whole slide images.
* The dataset used for evaluation is comparably large, as it contains 667 whole slide images.
* The evaluation is comparably comprehensive.

**Weaknesses:**

* The related work is not thorough enough. Better to discuss related works about gene expression prediction based on histopathology slides, as this work belongs to this category.
* The technical novelty is limited, as it just directly applies existing models (e.g. Attention-Based MIL) to a specific clinical problem.  As we know, it is common to apply MIL on histopathology WSI analysis. From the perspective of technique, there is no significant difference to previous works.


**Deanonymize Review:**

no

**Final Rating After The Rebuttal:**

4: Weak Accept

**Justification Of The Final Rating:**

Thank authors for the reply and the modification of the manuscript.  I am satisfied with the answers about the implementation. I agree with other reviewers about the clinical contribution of this work, so I upgrade the score to 'weak accept'.

**Paper Type:**

validation/application paper

**Questions To Address In The Rebuttal:**

* what's the threshold used to determine the accuracy ?
* For the ResNet-50 pretrained on ImageNet, it is that the backbone remains fixed during multiple instance learning training ?
* There are typos in the learning rate and weight decay.
* How many patches are cropped from each slide ?
* Where do you apply the dropout operation ?

**Special Issue:**

no

---

### Official Review · Reviewer_AUDK · 2022-01-23

**Confidence:** 4
**Preliminary Rating:** 4
**Recommendation:** Poster

**Summary:**

In this paper, the authors present a multiple instance learning approach to classify H&E-based WSIs of patients diagnosed with primary melanomas into a prognostic biomarker usually derived from transcriptomic data ("high-", "intermediate", "low-immune"). The authors use existing methods but systematically investigate meaningful combinations of the importance of pre-trained models and attention mechanisms. The results indicate relatively high performance at the task at hand when simplified to only "low-" and "high-immune". The study design and interpretation of the results are solid and valid.

**Strengths:**

Overall, the motivation for setting up this study is well-described, and the methods investigated are all relevant for the clinical task, which makes it easy for the reader to read. Even though the novelty of methods is not high, the clinical application area combined with the studied methods and dataset makes it a valuable contribution to the field. There is a good systematic design of the experiments (combination of known methods), which is sound for the application field. All relevant related works are introduced well and used throughout for making choices with good argumentation.

**Weaknesses:**

Few weaknesses in this paper can be addressed by the author:
- The explanation of the immunological groups is not clear in terms of why these require slide-level ground truth and cannot be derived from simpler models targeting the underlying biomarker, e.g., directly targeting TIL infiltration.
- The first results with "low-", "intermediate-", and "high immune" groups are considered "quite low", and then the authors simplify the problem by excluding the "intermediate" class. However, there is no explanation of why this is OK to do, i.e., the clinical impact of doing so. Furthermore, there are no considerations provided by the authors on how these should be handled in a testing scenario, or how the model can be utilized to discriminate between this group (see e.g., "Improved breast cancer histological grading using deep learning" Wang et al. (2022) for inspiration)
- The authors show that in some settings, the SSL approach trained on pathology images is superior to ImageNetpre-trained feature extractors. However, the results seem to depend on magnification (ImageNet models are better at 40X and 10X), and the discussion of why this is, can be improved.
- The discussion on why some attention worked better than others could be stronger. The authors present results from relevant attention mechanisms but provide no results or discussion on why there is a difference between them.
- Training and system requirements are not described, which is important for reproducibility.


**Deanonymize Review:**

yes

**Detailed Comments:**

Overall, a well-written paper that only has minor things that easily be improved by the authors.
Are the code and dataset released publicly? If not, please state it.


**Final Rating After The Rebuttal:**

4: Weak Accept

**Justification Of The Final Rating:**

The authors did a good job answering the questions during the rebuttal stage, and the paper is stronger now. Thank you for your answers. The paper is sufficient to be presented as a poster at the conference. Make sure to be explicit how this paper is different than other SSL and MIL papers in the field.

**Paper Type:**

validation/application paper

**Questions To Address In The Rebuttal:**

The questions the authors should address:
- Why needed to train on slide-level "genomic" labels if the groups are high and low immune infiltration? Why are other model types not better suited for this task?
- How to handle the "immediate" group if excluded from the experimental setup?
- Why is SSL better at 20X while ImageNet-based models are better at 10X and 40X?


**Special Issue:**

no

---

### Official Review · Reviewer_Yvvh · 2022-01-25

**Confidence:** 3
**Preliminary Rating:** 4
**Recommendation:** Poster

**Summary:**

In this paper, the authors utilize techniques for whole slide image classification on the classification of immune subgroups derived from transcriptomic data. The approach consists of using an encoder backbone that compresses input patches into smaller feature embeddings, which are then forwarded into an (attention) decoder, that can weigh the importance of each patch, while maintaining a full field of view of the original image.

The results indicate that neural networks are able to divide the subgroups on a basic level (low vs high), although subtyping all three subgroups remains a difficult challenge.

**Strengths:**

* The paper is well-written. Their intended experimental design is clear and easy to follow. They use the most recent advances in WSI classification such as CLAM, and address one of its primary shortcomings of using pre-trained ImageNet embeddings, by replacing it with an encoder that was trained on histopathology using self supervised learning (SSL ResNet18). All in all, the paper is a sound combination of existing techniques.

**Weaknesses:**

* Unclear comparisons between model designs. This is caused by the fact that they use two different ResNets with different pre-trained embeddings (see Questions to address during the rebuttal).

* It remains unclear whether subtyping all groups (low, intermediate, high) is possible. The results in table 1 only provide weak evidence. The results in table 2 indicate more favorable performance, but are based on exluding the intermediate group.
An explanation is missing on how this can be overcome, and what is needed to improve the performance of the model when all 3 groups are considered.

**Deanonymize Review:**

no

**Detailed Comments:**

* The following statement can be found in the disccusion and conclusions:
 Furthermore, we found improved model performance by focusing on only the “high” and “low” immune subgroups.

   This statement needs to be revised or deleted: it is not the model that has increased in performance, but the problem has become   easier, which is also stated in the next sentence.

* The authors note that the SSL ResNet18 models have higher attention values in the darker tumour regions of the WSI. Is there an explanation for this? Could there be a bias for darker colours?

* Figure 2: it is unclear which model was used to create the attention maps: is it with, or without gating?

* Table 1 states both accuracy and AUC, but table 2 only displays AUC. I think that for completeness accuracies should be included.

* A section on the availability of the transcriptomic data is in the appendix, but are the whole slide images and codes also available somewhere?

Small typos:
4.2 model architectures and training:
Learning rate: 2x10^4, I think it should be 2x10e-4
Weight decay: 1x10^5, I think it should be 1x10e-5

6.Discussion and conclusion:
Final phrase: should be “a helpful” instead of “an helpful”

**Final Rating After The Rebuttal:**

4: Weak Accept

**Justification Of The Final Rating:**

The authors have given satisfactory answers to the questions regarding the chosen architectures, implementation choices, and clinical relevance. I recommend the paper to be accepted as a poster presentation.

**Paper Type:**

validation/application paper

**Questions To Address In The Rebuttal:**

1. Why is a ResNet50 feature extractor with pre-trained ImageNet weights compared to a SSL ResNet18 feature extractor with hispathology embeddings? Would it not have made more sense to use the same architecture twice (e.g. both ResNet18)? This would make the comparison between designs easier.

2. The proposed WSL method is able to divide immunity subgroups when there is a clear separation between the classes (low/high immunity groups only). However, when all subgroups are taken into account, the results are quite low, e.g. due to heterogeneity in the intermediate class. What could be done to potentially overcome this problem? As this seems to be a relevant challenge to solve before considering clinical applications.

**Special Issue:**

no

---

### Meta-Review · Area_Chair_MpQQ · 2022-02-20

**Recommendation:** Accept (Poster)
**Confidence:** 4

**Metareview:**

All reviewers agree that this paper has value in that it introduces SSL in a framework like CLAM, and achieves promising results in the context of a relevant clinical application. Therefore, I recommend acceptance as a poster presentation, as suggested by several reviewers.

---

### Decision · Program_Chairs · 2022-02-28

Accept